# Genome Wide Analysis of U-Box E3 Ubiquitin Ligases in Wheat (*Triticum aestivum* L.)

**DOI:** 10.3390/ijms22052699

**Published:** 2021-03-07

**Authors:** Dae Yeon Kim, Yong Jin Lee, Min Jeong Hong, Jae Ho Kim, Yong Weon Seo

**Affiliations:** 1Institute of Animal Molecular Biotechnology, Korea University, 145 Anam-ro, Seongbuk-Gu, Seoul 02841, Korea; dykim@korea.ac.kr; 2Division of Biotechnology, Korea University, 145 Anam-ro, Seongbuk-Gu, Seoul 02841, Korea; leeyj83@korea.ac.kr (Y.J.L.); jhkim169@korea.ac.kr (J.H.K.); 3Advanced Radiation Technology Institute, Korea Atomic Energy Research Institute, 29 Geumgu, Jeongeup 56212, Korea; hongmj@kaeri.re.kr

**Keywords:** U-box E3 ligase, wheat, developmental stages, abiotic stress

## Abstract

U-box E3 ligase genes play specific roles in protein degradation by post-translational modification in plant signaling pathways, developmental stages, and stress responses; however, little is known about U-box E3 genes in wheat. We identified 213 U-box E3 genes in wheat based on U-box and other functional domains in their genome sequences. The U-box E3 genes were distributed among 21 chromosomes and most showed high sequence homology with homoeologous U-box E3 genes. Synteny analysis of wheat U-box E3 genes was conducted with other plant species such as *Brachypodium distachyon*, barley, rice, *Triricum uratu,* and *Aegilops tauschii*. A total of 209 RNA-seq samples representing 22 tissue types, from grain, root, leaf, and spike samples across multiple time points, were analyzed for clustering of U-box E3 gene expression during developmental stages, and the genes responded differently in various tissues and developmental stages. In addition, expression analysis of U-box E3 genes under abiotic stress, including drought, heat, and both heat and drought, and cold conditions, was conducted to provide information on U-box E3 gene expression under specific stress conditions. This analysis of U-box E3 genes could provide valuable information to elucidate biological functions for a better understanding of U-box E3 genes in wheat.

## 1. Introduction

The ubiquitin–proteasome system (UPS), which regulates selective protein degradation via the 26S proteasome, is one of the major mechanisms for post-translational regulation of gene expression and protein quality control in eukaryotes. The UPS plays a significant role in the regulation of signal transduction, metabolic regulation, differentiation, cell cycle transition, and stress response by causing the degradation of specific proteins [1,2]. The UPS involves a cascade of three steps: ATP-dependent activation of ubiquitin by a ubiquitin-activating enzyme (E1), transfer of ubiquitin to a conjugating enzyme (E2), and conveyance of ubiquitin to a substrate protein by E3 [3].

The E3 ubiquitin ligases are the largest family among all three enzymes and are classified into different families based on their structure, function, and substrate specificity. Several types of E3 ubiquitin ligases are characterized by the presence of specific domains, including homology to the E6-AP C-terminus (HECT) domain and a Really Interesting New Gene (RING)/U-box E3 domain, and Cullin-RING ubiquitin ligase. The RING/U-box E3 ligase can act as a single component and transfer ubiquitin to a target protein directly, whereas Cullin-RING ubiquitin ligases (CRLs) are multi-component and work with the Skp1–Cullin–F-box (SCF) complex [4]. They mediate the transfer of the ubiquitin protein to the substrate directly or by generating an intermediate complex (E3 and ubiquitin) [5,6].

The U-box E3 gene is a 75 amino acid domain that was first identified in UFD2, the yeast E4 enzyme [7]. The U-box is a derived version of the RING-finger domain that lacks the hallmark metal-chelating residues. Most of the signature cysteines of the RING-finger are not conserved in the U-box [8], but it is likely to function similarly to the RING-finger in mediating ubiquitin conjugation of protein substrates [9]. Various kinds of U-box E3 gene families have been identified in different plant genomes. At least 66, 77, 125, and 67 U-box E3 genes have been reported in *Arabidopsis thaliana* [10], *Oryza sativa* [11], *Glycine max* [12], and *Hordeum vulgare* [13], respectively. The appearance of high numbers of U-box E3 proteins in plant genomes suggests that U-box E3 proteins could regulate various substrates and play diverse roles in post-translational modifications in many biological processes such as cellular oxidative stress during seedling establishment [14], plant organ development [15], circadian rhythm [16], leaf senescence [17], plant cell death, and defense [18]. In addition, U-box E3 proteins are involved in hormone and (a)biotic stress signaling with multiple regulations in plants [16,19,20,21,22]. Although it is known that the UPS participates in various plant developmental, (a)biotic, and hormone pathways, U-box E3 gene family members in wheat have not been elucidated yet.

Wheat is allohexaploid (2n = 6x = 42; AABBDD) with a huge genome of ~17 Gbp, which comprises three closely related homoeologous subgenomes consisting of a high percentage of repetitive sequences and homoeologous DNA copies. Bread wheat is not only an important cereal crop but also a model for the study of an allopolyploid plant with a large and highly repetitive genome. Recently, the reference sequence of a bread wheat cultivar, Chinese Spring, was released publicly and is available with gene annotations [23]. In this study, U-box E3 genes were identified and classified based on a hidden Markov model (HMM) search using the wheat reference genome. A phylogenetic tree was constructed, and chromosomal locations were mapped for the 213 genes that encode U-box E3 proteins. The synteny between wheat U-box E3 proteins and those of the progenitors of wheat (*Triticum urartu* and *Aegilops tauschii*) as well as *Horedeum vulgare*, *Oryza sativa*, and *Brachypodium distachyon*, was analyzed. Additionally, expression profiling of U-box E3 genes was conducted with RNA sequencing data for wheat at various developmental stages and under abiotic stresses such as heat, drought, and cold stress conditions. The identification of the U-box E3 gene family members in wheat will help understand the fundamental aspects of the ubiquitin proteasomal degradation system regulated by U-box E3 genes in plants. This study will provide novel information regarding the classification and expression of U-box E3 proteins that may be useful for agricultural approaches to ubiquitin proteasomal degradation in wheat.

## 2. Results

### 2.1. Identification of U-Box E3 Genes in the Wheat Genome

A total of 213 putative U-box E3 genes were identified by HMM profiling against the local protein database of the IWGSC (International Wheat Genome Sequencing Consortium) Reference Sequence v1.0 using HMMER3. These genes were validated by SMART (Simple Modular Architecture Research Tool) and Pfam (Protein Families Database) analysis to eliminate U-box E3 genes with redundant sequences showing low E-values and alignment scores, and 213 U-box E3 genes with the U-box domain were selected for further analysis (Appendix A). The U-box E3 genes were classified into seven sub-groups that contained U-box domains and other specific domains. Seventy-nine U-box genes possessing the armadillo (Arm) repeat domain were identified; this was the most abundant sub-group. Fifty-one U-box E3 genes contained only the U-box domain in their peptide sequence. The U-box domain and one or more specific domains, such as Usp (ubiquitin-specific protease) + protein kinase (PKinase), PKinase, tetratricopeptide repeat (TPR), and WD40 repeat domains, were identified in 36, 13, 9, and 9 U-box E3 genes, respectively. Sixteen U-box E3 genes contained a small number of other specific domains (UBO) such as protealysin propeptide (PLN_propep), ubiquitin elongating factor core (Ufd2P_core), zinc-binding domain (zf-CCHC_2), cyclophilin-type peptidyl-prolyl cis-trans isomerase (Pro_isomerase), and Usp domains (Figure 1 and Appendix A).

### 2.2. Gene Ontology Analysis of U-Box E3 Genes in the Wheat Genome

Gene ontology (GO) analysis of wheat U-box E3 genes was conducted using Blast2GO to predict the biological and molecular functions in cellular metabolism. Of the 213 U-box E3 genes, 201 were assigned the GO categories “biological process” and “molecular function” (Figure 2 and Appendix A). Within the “biological process” category, 194 U-box E3 genes (96.52%) were mapped to the GO terms “protein modification by small protein conjugation or removal” (GO:0070647) and “protein ubiquitination” (GO:0016567) in levels 8 and 9, respectively. Additionally, 194 U-box E3 genes (96.52%) were mapped to the GO terms “ubiquitin-like protein transferase activity” (GO:0019787) and “ubiquitin-protein transferase activity” (GO:0004842) in levels 4 and 5 of the “molecular function” category, respectively.

### 2.3. Phylogenetic Analysis and Chromosomal Location of U-Box E3 Genes

A phylogenetic analysis was conducted using the full-length protein sequence encoded by 213 U-box E3 genes provided by IWGSC Reference Sequence v1.0. The phylogenetic analysis of U-box E3 proteins possessed one or more functional domains as shown in Figure 3. As shown in Figure 1, U-box E3 genes containing only the U-box domain, referred to as U-box, grouped together. In addition, U-box E3 genes containing other specific functional domains, such as Arm, Ups + PKinase, PKinase, TPR, and WD40, were also clustered together and showed similar genetic structures in each group. The UBO sub-group (PLN_propep, Ufd2P_core, zf-CCHC_2, Pro_isomerase, and Usp) showed a scattered pattern. The chromosome locations of U-box E3 genes were identified using IWGSC Reference Sequence v1.0, except TraesCSU02G03640050, which does not define chromosomal location. Fifty, 45, and 34 U-box E3 genes were located on chromosomes 6, 7, and 2, respectively (Figure 4). These chromosomes possessed more than 60% (129 out of 212) of the wheat U-box E3 genes. Meanwhile, the number of U-box E3 genes on chromosomes 1 and 3 was determined to be 15 and 16, respectively. Seventy-one, 69, and 72 U-box E3 genes were located in the A, B, and D genomes of wheat homoeologous chromosomes.

### 2.4. Detection of Homoeologous U-Box E3 Genes with High Sequence Homology between Triticum uratu, Aegilops tauschii, and Wheat

To detect U-box E3 gene pairs with high sequence homology, U-box E3 genes that matched using BLASTP with strict parameters, such as an *E*-value < 1 × 10^−10^, minimum sequence identity > 90%, and bitscore > 500, were selected and used for constructing synteny maps of wheat U-box E3 genes (Figure 5A). A total of 73 major types of homoeologous U-box E3 genes, including 1:1:1, 1:1:0, 0:1:1, and 1:0:1 types, were detected in the wheat genome. The most abundant type of homoeologous U-box E3 genes, with 52 pairs, was the 1:1:1 type, which means that one U-box E3 gene was located in each homoeologous A, B, and D genome (Appendix A). The 1:1:0, 1:0:1, and 0:1:1 homoeologous pair types were detected 3, 6, and 12 times, respectively, and one 1:3:1 and one 1:1:2 homoeologous pair type were also found in the wheat genome. In addition, U-box E3 genes in the diploid progenitors and ancestors of wheat, *T. uratu* and *A. tauschii*, were identified to investigate evolutionary changes in U-box E3 genes from diploid to hexaploid genomes (Figure 5A). A total of 54 and 70 orthologous pairs of wheat U-box E3 genes were detected in *T. uratu* and *A. tauschii*, respectively. In *T. uratu*, they consisted of 4, 10, 3, 2, 12, 13, and 10 pairs on chromosomes 1A to 7A, respectively. In *A. tauschii*, they consisted of 4, 11, 6, 8, 8, 18, and 17 pairs on chromosomes 1D to 7D, respectively. Most of the U-box E3 sub-groups containing specific domains mapped to *T. uratu* and *A. tauschii* were the Arm, U-box E3, and Usp + PKinase sub-groups. There were 14 (25.9%), 13 (24.1%), and 12 (22.2%) orthologous gene pairs between wheat and *T. uratu* in the Arm, Usp + PKinase, and U-box E3 sub-groups, respectively. The Arm (37.1%), U-box E3 (27.1%), and Usp + PKinase (15.7%) sub-groups showed a high proportion of orthologous gene pairs between wheat and *A. tauschii* (Appendix A). In addition, 18 pairs of duplicated U-box E3 genes located in non-homologous chromosomes were detected in the wheat genome (Figure 5B and Appendix A). Six pairs of U-box E3 genes showed duplication events in the equal chromosome on 4A, 5A, 5B, and 6B. In addition, 12 pairs of segmental duplication events were detected between different chromosomes, such as between chromosomes 4A and 7A/7D, 4B/4D and 5A, and 5A/5B/5D and 6A. Interestingly, duplication events were observed on chromosomes 4 to 7 but not on chromosomes 1 to 3. The number of U-box E3 genes on chromosomes 4 and 5 were not only lower than those on chromosomes 6 and 7 (Figure 4), but a high number of duplication events occurred on chromosomes 4 and 5, indicating that the duplication events could play important roles in the extension of each chromosome of the U-box E3 genes in the wheat genome.

### 2.5. Synteny Analysis of U-Box E3 Genes in Wheat and Other Plants

Next, comparative synteny maps were constructed with wheat and other plants, such as *Brachypodium*, barley, and rice, to investigate the evolutionary history of wheat U-box E3 genes. One hundred and seventy-five wheat U-box E3 genes were matched with 43 *Brachypodium* U-box E3 genes, consisting of 13, 7, 15, 5, and 3 U-box E3 genes on chromosomes 1 to 5, respectively (Figure 6A and Appendix A). Interestingly, wheat U-box E3 genes were concentrated in specific *Brachypodium* chromosomes; 12 out of 16 (75.0%), 12 out of 12 (100%), 21 out of 27 (77.8%), 15 out of 17 (88.2%), 19 out of 21 (90.0%), 48 out of 50 (96.0%), and 20 out of 32 (62.5%) genes on wheat chromosomes 1 to 7 were mapped to 4, 4, 6, 6, 9, 16, and 7 U-box E3 genes on chromosomes 2, 5, 2, 1, 4, 3, and 1 of the *Brachypodium* genome. In the case of the barley genome, 207 wheat U-box E3 genes were matched to 65 barley U-box E3 genes, consisting of 5, 10, 6, 6, 9, 15, and 14 U-box E3 genes on barley chromosomes 1 to 7, respectively (Figure 6B and Appendix A). All of the U-box E3 genes on wheat chromosomes 1, 2, 3, and 7 were mapped to barley U-box E3 genes on identical barely chromosome numbers, and 14 out of 21, 33 out of 39, and 47 out of 41 U-box E3 genes on chromosomes 4, 5, and 6, respectively, were also mapped to barley U-box E3 genes on the same barley chromosome numbers. For rice, 75 wheat U-box E3 genes were matched to 25 syntenic rice U-box E3 genes (Figure 6C and Appendix A). The rice U-box E3 genes showed high sequence similarity with wheat U-box E3 genes that were located on rice chromosomes, except for chromosomes 7 and 11.

### 2.6. Expression Analysis of U-Box E3 Genes

To investigate the expression pattern of U-box E3 genes in wheat developmental stages and response to abiotic stress, we collected and analyzed RNA sequencing data from expVIP (Figure 7 and Figure 8). A total of 209 RNA-seq samples representing 22 tissue types, from grain, root, leaf, and spike samples across multiple time points, were analyzed and used to observe the expression patterns of U-box E3 genes in different tissues and developmental stages (Appendix A). The TPM value of U-box E3 genes was used for clustering of expression patterns by k-means methods and grouped into six clusters based on their TPM values (Figure 7 and Appendix A). The 43 U-box E3 genes in Group 1 showed high expression levels in the shoot apical meristems and shoot axis in the vegetative stage and stigma, ovary, and spikes in the reproductive stage. The 69 U-box E3 genes in Group 2, which contained the most abundant genes, showed leaf-specific expression and were highly expressed in leaves, such as the first leaf sheath and blade in the vegetative stage, and the flag leaf sheath and flag leaf blade in the reproductive stage. Groups 3 and 4 contained 33 and 21 U-box E3 genes, respectively, with genes showing high expression patterns in floral organs, including the anthers, glumes, and lemma in the vegetative stages. The expression of 33 and 32 U-box E3 genes in Groups 5 and 6 was mainly detected in roots, such as the radicles, root apical meristems, and roots in the vegetative stage. The 34 U-box E3 genes in Group 7 did not show tissue- or developmental stage-specific patterns and were expressed in all developmental stages of wheat.

In Group 2 of Figure 4, TraesCS2A02G079300.1, TraesCS4A02G498700.1, and TraesCS5A02G542700.1 were validated by qRT-PCR, and the results corroborated the RNA-Seq results that showed developmental stage-specific expression in leaves in the vegetative stage (Figure 9A–C). Conversely, TraesCS6D02G022900.1 and TraesCS7D02G303700.1 in Group 2 of Figure 7 were highly expressed in spikes in the reproductive stage (Figure 9D,E).

Next, we analyzed the expression of U-box E3 genes under abiotic stress conditions, including drought, heat, and cold stress. A total of 118 genes showed drought- and heat-stress-specific expression patterns and were divided into seven groups by the k-means clustering method (Figure 8A and Appendix A). Forty-eight and 12 wheat U-box E3 genes of Groups 1 and 2 showed drought-specific expression patterns. Conversely, the expression of 22 and 10 U-box E3 genes of Groups 3 and 4 was mainly induced under heat stress conditions ten and nine U-box E3 genes of Groups 5 and 6 showed increased expression under heat stress and under combined heat and drought stress (Figure 8A). Consistent with these results, TraesCS2D02G277300.1, TraesCS2B02G499300.1, TraesCS5A02G198800.1, TraesCS6A02G419200.1, and TraesCS6D02G407400.1 in Group 2 of Figure 8A were upregulated under drought stress (determined using qRT-PCR) (Figure 9F–J). In the case of cold stress, 36 of the 64 U-box E3 genes were downregulated, whereas 28 U-box E3 genes were upregulated under cold stress (Figure 8B and Appendix A). Interestingly, 17 of the 28 upregulated U-box E3 genes and 20 of the 38 downregulated U-box E3 genes under cold stress were highly expressed under both drought and heat stress. TraesCS2A02G276400.1, TraesCS3A02G394600.1, TraesCS2D02G275400.1, TraesCS2B02G294100.1, TraesCS2D02G456500.1, TraesCS3B02G426600.1, TraesCS6B02G268200.2, TraesCS6D02G274500.1, TraesCS6D02G292200.1, TraesCS3D02G388400.1, TraesCS6B02G222200.1, TraesCS2A02G456200.2, and TraesCS7A02G194900.1 were upregulated under drought and cold stress conditions, and TraesCS5B02G200100.1 and TraesCS6B02G027000.1 were upregulated under heat and cold stress conditions. TraesCS4D02G337200.1 and TraesCS7B02G194100.1 were upregulated under heat stress, combined heat and drought stress, and cold stress conditions.

## 3. Discussion

E3 ubiquitin ligase is the largest family of enzymes that catalyze the covalent attachment of a small protein modifier, ubiquitin, to substrates in eukaryotic cells, causing ubiquitin proteasomal degradation [24]. The U-box E3 proteins play a regulatory role in protein degradation through the proteolytic mechanism of multi-protein E3 ubiquitin ligase in response to cellular signals during plant development and growth, hormone response, and stress responses [25,26,27,28]. In wheat, *TaPUB1* improved plant salt and drought stress tolerance [29], and *TaPUB4* was involved in the regulation of pollen development by regulating sucrose and starch metabolism in anthers. Another wheat U-box E3 protein, *TaPUB15*, was induced by salt, abscisic acid, temperature of 4 °C, and polyethylene glycol (PEG). Although U-box E3 genes have important roles in plant development and various stress responses, to date, the functional analysis of wheat U-box E3 genes has proceeded at a single-gene level.

In the present study, we identified 213 U-box E3 ubiquitin ligases in the wheat genome by HMMER analysis using the Pfam and InterPro databases. A large number of wheat U-box E3 genes (79, 37%) contained an Arm repeat domain, which is an approximately 40 amino acid-long tandem repeat motif (Figure 1 and Figure 3). The U-box domain is found in combinations with various additional domains, of which ARM repeats are also the most common in Arabidopsis and rice [11,30]. The interaction with the ARM repeats does not necessarily result in ubiquitination of the interactors to the E2–ubiquitin conjugate; they bind protein kinase, which is an essential component of the signaling pathway and crosstalk between ubiquitination and phosphorylation [31]. For example, AtPUB-ARM E3 ubiquitin ligase interacts with S-domain receptor kinases involved in ABA signaling [19], and AtPUB22 interacts with MITOGEN-ACTIVATED PROTEIN KINASE3 (MPK3) to control the immune response [32]. Fifty-three U-box E3 genes contained only the U-box domain in their peptide sequence, and a small number of U-box E3 genes with Usp + PKinase, PKinase, TPR, and WD40 repeat domains were detected in the wheat genome. The existence of various types of U-box E3 gene families indicates that they are involved in diverse biological functions [33].

Wheat is allohexaploid (2n = 6x = 42; AABBDD) with a huge genome of ~17 Gbp that consists of three closely related homoeologous subgenomes. The wheat genome contains a high percentage of repetitive sequences and homoeologous DNA copies from the three subgenomes [34]. A total of 208, 179, and 219 pairs of U-box E3 genes were matched by BLASTP with strict parameters (*E*-value < 1e^−10^, minimum sequence identity > 90%, and bitscore > 500) and were used to construct synteny maps of wheat, *T. uratu*, and *A. tauschii*, respectively. The 208 wheat U-box E3 pairs consisted of 21 pairs, 52 triplets, and two more quadruplet matches, indicating that three homoeologous chromosomes have a high percentage of sequence homology and closely related genome sequences (Figure 5A and Appendix A). In the identification of U-box E3 genes in wheat and its ancestors, 54 and 70 orthologous pairs of wheat U-box E3 genes were detected in *T. uratu* and *A. tauschii*, respectively. The number of U-box E3 genes in subgenomes A and D increased in transition from diploidy to hexaploidy as shown by 50 in *T. uratu* to 71 in the A genome and 67 in *A. tauschii* to 72 in the D genome (Figure 5A and Appendix A). These results indicate that dynamic gene gains occur broadly during the formation of hexaploidy [34]. We investigated the occurrences of putative gene duplication of wheat U-box E3 genes (Figure 5B and Appendix A). The duplicated U-box E3 genes were detected by identification of paralogous relationships by sequence similarity [35]. As the U-box E3 genes on each homoeologous chromosome showed high sequence homology and similarity, gene duplication events of U-box E3 genes between non-homoeologous chromosomes in the wheat hexaploid genome were investigated by BLASTP based on the following criteria: (a) *E*-value threshold < 1e^−10^; (b) the alignment covered > 80% of the longer gene; and (c) the aligned region had an identity > 90%. Six pairs of tandem duplication events adjacent to each other on the same chromosome and 12 pairs of segmental duplication events among various chromosomes were identified, suggesting that the duplication events of U-box E3 genes could play an important role in the functional extension and diversity of U-box E3 genes in the wheat genome. Synteny analysis with other plants identified 175, 207, and 75 U-box E3 genes in wheat that were closely related to 65 barley U-box E3 genes, 43 *B. distachyon* U-box E3 genes, and 25 rice U-box E3 genes, respectively (Figure 6 and Appendix A). The number of U-box E3 genes in barley, A, B, and D genomes were 65, 71, 69, and 72, respectively, and the number of U-box E3 genes in barley was similar to that in wheat. In a previous study, a high degree of sequence similarity was confirmed between wheat and barley, comparing transcript map data to sequenced model genomes [36], as the divergence of wheat and barley occurred approximately 11.6 million years ago [37]. The phylogenetic position of *B. distachyon* indicates that it is suitable for use as a representative grass species with a large genome, including for cool-season grasses such as wheat, barley, and rice [38]. Rice has also been an ideal grass model for several decades [39]. The orthologs of *B. distachyon* and rice U-box E3 genes can be used for molecular and functional analyses to reveal the roles of U-box E3 genes in crucial agronomic crop traits.

Many studies have revealed that plant U-box E3 genes play important roles in the regulation of plant development. In the present study, 43 U-box E3 genes in Group 1 showed high expression levels in shoot apical meristems in the vegetative stage and stigma, ovaries, and spikes in the reproductive stage. The expressions of 33 and 32 U-box E3 genes in Groups 5 and 6 were mainly detected in roots, such as the radicle and root apical meristem, and roots in the vegetative stage. Thirty-three and 21 U-box E3 genes were in Groups 3 and 4 and showed high expression level patterns in floral organs, including the anthers, glumes, and lemma in the vegetative stage (Figure 7 and Appendix A). Similarly, the *pub4* mutants in Arabidopsis showed higher levels of cell proliferation and division in root and shoot apical meristems. The root apical meristem of *pub4* exhibited a decrease in inhibition of root cell proliferation and stem cell maintenance by CLAVATA3 (CLV3) [25]. The size of SAM was dramatically increased in the double mutants (*clv2/pub4* and *clv3/pub4*), and the number of carpels was increased in the double mutants (*clv2/pub4* and *clv3/pub4*) [40]. The *pub4* mutant and the *pub2/pub4* double mutant exhibited defects in stamen development, tapetum development, and male fertility [41]. The plant U-box E3 genes are affected by phytohormones from cell division and differentiation in the apical meristem to flowering and maturation throughout the wheat developmental stages. The *pub2* and *pub4* plant seedlings were insensitive to cytokinin, and *pub10* plants were sensitive to jasmonic acid, which inhibited root growth [26,42]. In addition, *pub9-1* reduced root growth and lateral root formation by altering auxin response [42]. Flowering time was increased in *pub12/pub13* double mutants in Arabidopsis, whereas the mutant rice U-box E3 protein SPOTTED LEAF11 (SPL11) delayed flowering time in rice [43,44]. In addition, 34 U-box E3 genes in Group 7 were detected and expressed in the overall developmental stages of wheat. Therefore, the expression profiling of U-box E3 genes in various tissues and developmental stages could indicate that some U-box E3 genes are expressed in a tissue- and developmental stage-specific manner and have important functions specific to the tissue and stage type, whereas the ubiquitously expressed F-box genes are involved in general cellular machinery. Many studies have reported that U-box E3 genes respond to abiotic stress, such as light, drought, and salt, in other plants [27,45,46,47,48]. In the present study, 60 and 51 genes exhibited specific responses to drought and heat stress conditions, respectively (Figure 8A). Interestingly, 17 out of 28 U-box E3 genes that were upregulated and 20 out of 38 U-box E3 genes that were downregulated under cold stress conditions were also found in one of the groups in Figure 8A, which were highly expressed under drought and heat stress conditions. These genes could be considered multi-stress genes. Their molecular functions and responsive mechanisms need to be clarified in further studies.

In this study, we identified and characterized U-box E3 genes in wheat for the first time. A total of 213 U-box E3 genes were identified through genome-wide analysis and clustered by functional domains in each U-box E3 gene. Most of the U-box E3 genes had high sequence homology with other U-box E3 genes located on homoeologous chromosomes, and duplication events were surveyed. Synteny analysis of U-box E3 genes was conducted between wheat and *Brachypodium*, barley, rice, and wheat progenitors such as *T. uratu* and *A. tauschii*. Expression profiles of wheat U-box E3 genes under various developmental stages, tissues, and abiotic stress conditions were analyzed, and some U-box E3 genes showed tissue-, developmental stage-, and abiotic stress-specific expression patterns. Our results provide valuable information for further functional analysis and could be useful to elucidate the mechanisms of essential agronomic traits in wheat.

## 4. Materials and Methods

### 4.1. Identification and Sequence Analysis of U-Box E3 Genes in the Wheat Genome

The IWGSC wheat reference sequence (IWGSC Reference Sequence v1.0; https://urgi.versailles.inra.fr/download/iwgsc/IWGSC_RefSeq_Annotations/v1.0/ (accessed on 17 August 2018)) was provided by Unité de Recherche Génomique Info (URGI, https://urgi.versailles.inra.fr/ (accessed on 17 August 2018)). HMM profiling of U-box E3 proteins was conducted with the HMM files of U-box (PF04564) domains, which were provided by Pfam [49] and searched against protein sequences of the wheat genome using HMMER3 with default parameters [50]. The U-box E3 genes were verified using SMART [51] with an *E*-value threshold of 1e^−5^ for significance. Additionally, Blast2GO was used for functional annotation of the U-box E3 genes [52]. Local BLASTP was conducted with the peptide sequence of the Poaceae family from the NCBI (National Center for Biotechnology Information) to initiate GO analysis for the construction of xml files (*E*-value < 1.0e^−3^). Then, mapping steps were executed to search GO terms and UniProt, followed by annotation steps that were executed with default parameters (*E*-value < 1.0e^–6^, annotation cutoff = 55, GO weight = 5).

### 4.2. Phylogenetic Analysis and Chromosomal Locations of U-Box E3 Genes

A phylogenetic tree file (ph file; Newick format) was generated using the neighbor-joining method in DoMosaics [53] with the protein sequences of the selected wheat U-box E3 genes for phylogenetic analysis. Bootstrapping was conducted with 1000 replications. The Interactive Tree of Life (iTOL) [54] was applied to construct a phylogenetic tree of the U-box E3 genes with the phylogenetic tree file analyzed by DoMosaics. Chromosomal locations of U-box E3 genes in the wheat genome were obtained from the GFF file of IWGSC Reference Sequence v1.0 and then plotted using MapChart 2.30 [55].

### 4.3. Genome Distribution with High Sequence Homology and Gene Duplication of U-Box E3 Genes

BLASTP was used to predict sequence homology of U-box E3 genes with an *E*-value < 1 × 10^−10^, sequence identity > 90%, and bitscore > 500. Gene duplication events between non-homoeologous chromosomes of U-box E3 genes in the wheat genome were investigated based on the following criteria: (a) the alignment covered >80% of the longer gene and (b) the aligned region had an identity > 90%. The U-box E3 genes showing high sequence homology and duplicated genes in the wheat genome were visualized by Circos-0.69 program [56].

### 4.4. Synteny Analysis of U-Box E3 Genes in Wheat and Other Plants

The locations of U-box E3 orthologous genes on the chromosomes of *T. urartu* and *A. tauschii*, which are the progenitors of the AA and DD genomes, respectively, were predicted from Ensembl Plants (http://plants.ensembl.org/ (accessed on 20 November 2017)), with an *E*-value threshold of 1 × 10^−10^ and sequence identity > 90%. The IWGSC Reference Sequence v1.0 provided by URGI, along with *B. distachyon*, *H. vulgare*, and *O. sativa* genome sequences, available from Ensembl Plants (http://plants.ensembl.org/ (accessed on 17 August 2018)), were downloaded and used for synteny analysis of U-box E3 genes. For synteny analysis of wheat against *Brachypodium*, barley, and rice, BLASTP was conducted with an *E*-value threshold of 1 × 10^−10^ and sequence identity > 80%. Intragenomic and intergenomic comparisons were conducted using Circos [56].

### 4.5. Expression Analysis of U-Box E3 Genes Using RNA-Seq Data

To analyze the expression of U-box E3 genes in different tissues and developmental stages, 290 RNA-sequencing samples representing 22 tissue types, from grain, root, leaf, and spike samples across multiple time points, were downloaded and analyzed [57]. In addition, 20 RNA-seq samples under abiotic stress conditions, such as heat, drought, and cold stress, were used to analyze the expression pattern of U-box E3 genes in response to stress. The RNA sequencing data (SRP045409) were used for the analysis of U-box E3 genes under drought and heat stress conditions. The SRP045409 consisted of wheat seedlings under normal conditions (22 °C/18 °C (day/night), 16 h/8 h (light/dark), and 50% humidity), 1 h and 6 h treatments of heat stress (40 °C), drought stress (20% (m/V) PEG-6000), and combined heat and drought stress (40 °C and 20% PEG-6000), respectively [58]. The SRP043554 was conducted with the samples belonging to wheat plants in the three-leaf stage under normal (grown at 23 °C for 4 weeks after germination) and cold stress (grown at 23 °C for 2 weeks followed by 4 °C for another 2 weeks) conditions. The log_2_ transformed transcripts per million (TPM) value was calculated and used to construct heatmaps of U-box E3 gene expressions at different developmental stages and stress conditions, and Mev software was used for k-means clustering of differentially expressed U-box E3 genes [59].

### 4.6. Plant Materials and Real-Time Quantitative RT-PCR

The seeds of common wheat (*Triticum aestivum* L.) cv. Keumgang (IT 213100) developed by the National Institute of Crop Science was vernalized at 4 °C for 4 weeks to synchronize growth and then the wheat seedlings were transferred to soil (Sunshine Mix 1, Sun Gro Horticulture, MA, USA) pots. The plants were grown in controlled environments at 23–26 °C under long-day conditions (16 h/8 h day/night). Using the Zadoks growth scale (Z) [60] as a reference, we collected the following plant samples at different developmental stages with three independent biological replicates: leaves at Z13 (three leaves emerged; Stage 1), leaves at Z24 (main stem and four tillers present; Stage 2), leaves at Z51 (leaf at the tip of ear just visible, booting stage; Stage 3), spikelets at Z61 (beginning of anthesis; Stage 4), and spikelets at Z73 (early milk development; Stage 5). For the drought stress treatment, wheat seeds were vernalized at 4 °C for 4 weeks to synchronize growth and then transferred to an Incu Tissue (72 × 72 × 22 mm; SPL Life Sciences, Gyeonggi-do, Korea) containing a polypropylene net floating on Hoagland solution (Sigma–Aldrich, St. Louis, MO, USA). Seedlings in the three-leaf stage on Hoagland solution were treated with 20% PEG6000 (Sigma–Aldrich, St. Louis, MO, USA) for simulation of drought stress. Samples were collected 6, 12, and 24 h after treatment with three independent biological replicates. Total RNA was extracted from three independent biological replicates at each developmental stage and under each stress condition using TRIzol reagent (Invitrogen, Waltham, MA, USA), and total RNA samples were treated with DNase I and cleaned to eliminate any contaminating genomic DNA. First-strand cDNA synthesis was conducted using a Power cDNA Synthesis kit (iNtRON Biotechnology, Gyeonggi-do, Korea) with 1 μg of total RNA and 2× SYBR premix Ex Taq II (Takara, Shiga, Japan) in a volume of 25 μL, including the first-strand cDNA, in an iCycleriQTM Real-Time PCR System (Bio-Rad, Hercules, CA, USA). Homoeological-specific regions of selected U-box E3 genes were used to design gene-specific primers for validation of qRT-PCR, and primers used in this study are shown in Appendix A.

## Figures and Tables

**Figure 1 ijms-22-02699-f001:**
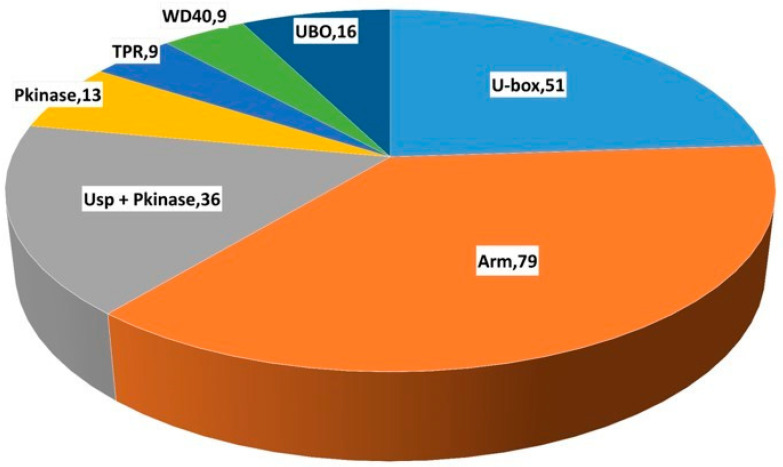
Classification of wheat U-box genes based on functional domains. The number and type of U-box E3 genes are shown. U-box: U-box E3 genes containing only the U-box domain; Arm: armadillo (Arm) repeat; Usp + PKinase: ubiquitin specific protease and protein kinase; PKinase: protein kinase; TPR: tetratricopeptide repeat; WD40: WD40 repeat; UBO: U-box containing small numbers of other specific domains such as protealysin propeptide (PLN_propep), ubiquitin elongating factor core (Ufd2P_core), zinc-binding domain (zf-CCHC_2), cyclophilin-type peptidyl-prolyl cis-trans isomerase (Pro_isomerase), and Usp.

**Figure 2 ijms-22-02699-f002:**
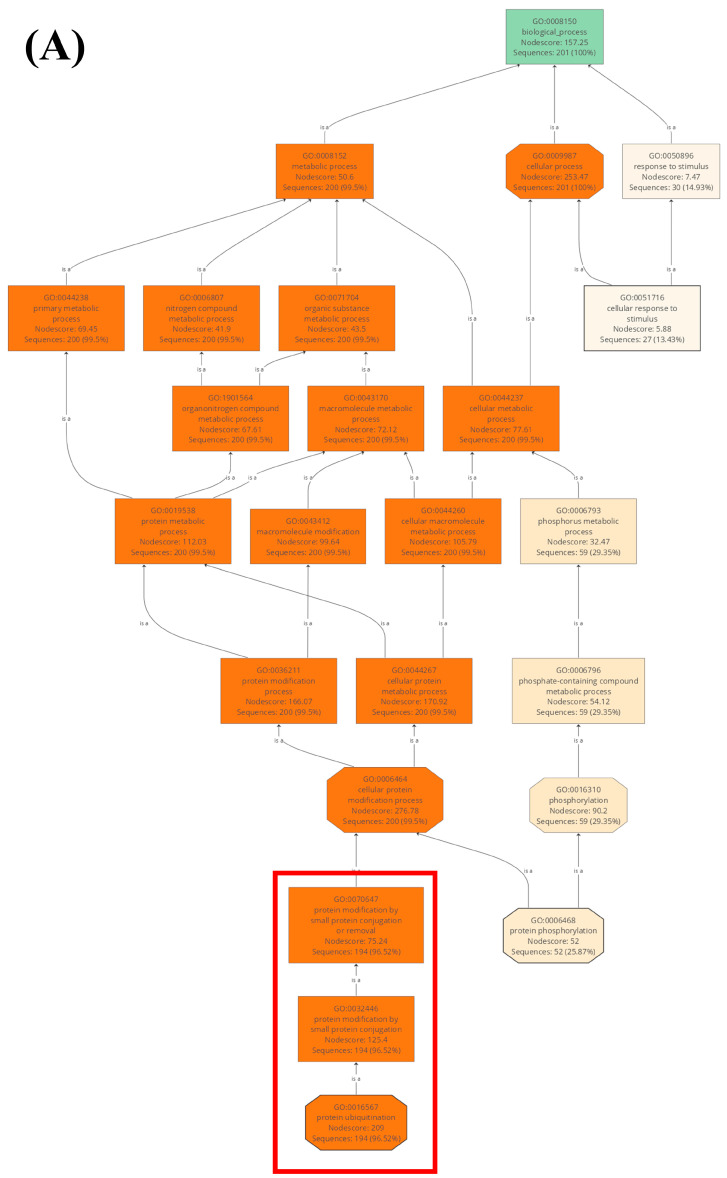
Gene ontology (GO) analysis of wheat U-box E3 genes. GO terms of U-box E3 genes were in the categories “biological process” (**A**) and “molecular function” (**B**). Red and blue boxes in (**A**,**B**) indicate that most of the selected U-box E3 genes have the function “protein ubiquitination” in biological process and molecular function, respectively.

**Figure 3 ijms-22-02699-f003:**
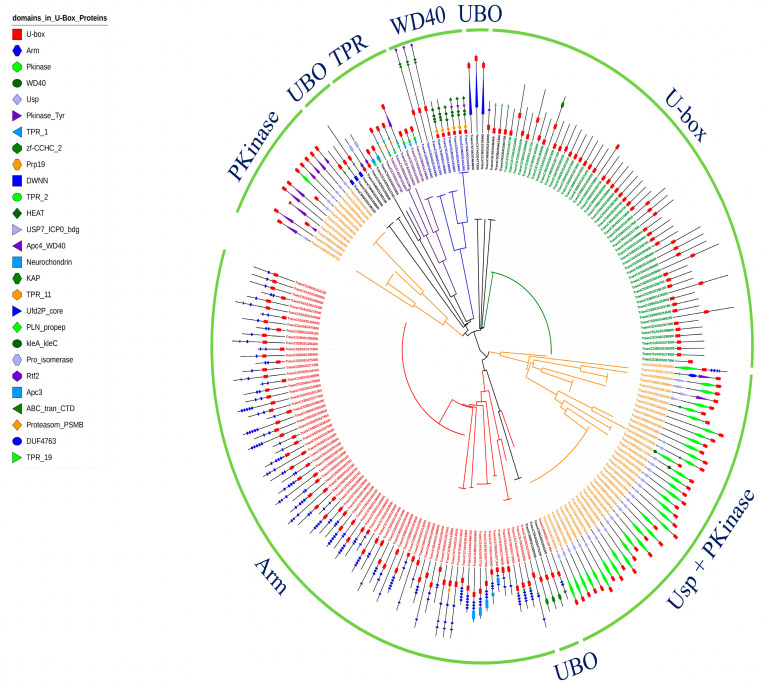
Phylogenic analysis of wheat U-box E3 genes. A phylogenetic tree was generated using the Interactive Tree of Life (iTOL) tool with the amino acid sequences of U-box E3 proteins that contained U-box domains and one or more known functional domains in the peptide sequences.

**Figure 4 ijms-22-02699-f004:**
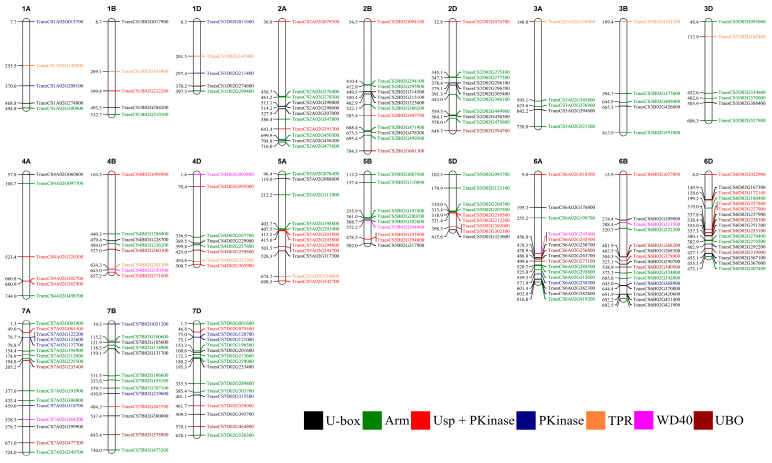
Chromosomal locations of wheat U-box E3 genes. The positions of U-box E3 genes (unit: Mb) are shown on the left of the chromosome, while the numbers on the right indicate the gene identifiers.

**Figure 5 ijms-22-02699-f005:**
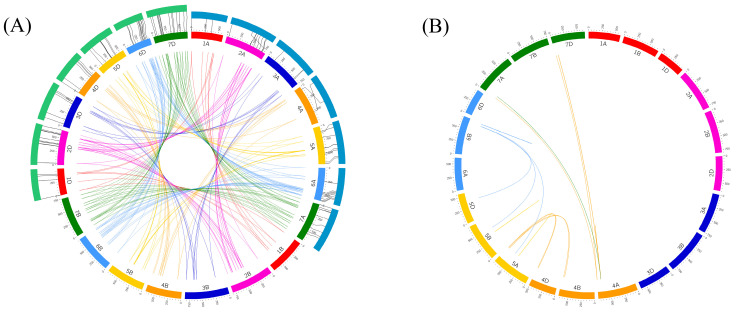
Synteny analysis and gene duplication of wheat U-box E3 genes using the BLASTP algorithm. (**A**) An *E*-value < 1e^−10^, minimum sequence identity > 90%, and bitscore > 500 were used to create a synteny map among wheat, *T. urartu,* and *A. tauschii*. The green annulus on the top left represents chromosomes of *A. tauschii*, and the blue annulus on the top right represents chromosomes of *T. urartu*. Homoeologous genes in each chromosome of wheat are linked by lines with identical color. (**B**) Duplicated U-box E3 gene pairs identified in wheat with the alignment covered >80% of the longer gene and the identity the aligned region > 90%. Duplicated gene pairs are depicted in corresponding colors and linked using lines with the corresponding color.

**Figure 6 ijms-22-02699-f006:**
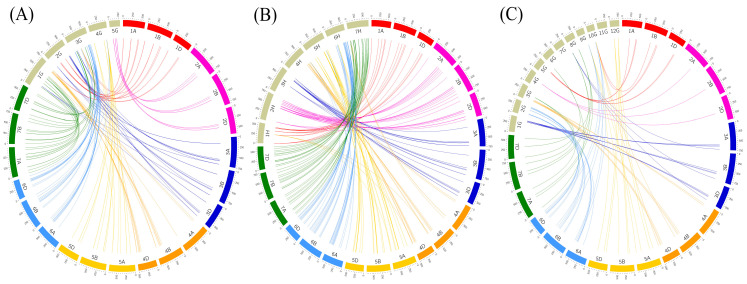
Synteny analysis of wheat U-box E3 genes with (**A**) *Brachypodium*, (**B**) barley, and (**C**) rice. For synteny analysis of wheat against *Brachypodium* and rice, BLASTP was conducted with an E-value threshold of 1 × 10^−10^ and sequence identity > 80%. Colored lines denote syntenic regions between wheat chromosomes and other plants. The genome size of *Brachypodium* and rice was enlarged by a factor of 10.

**Figure 7 ijms-22-02699-f007:**
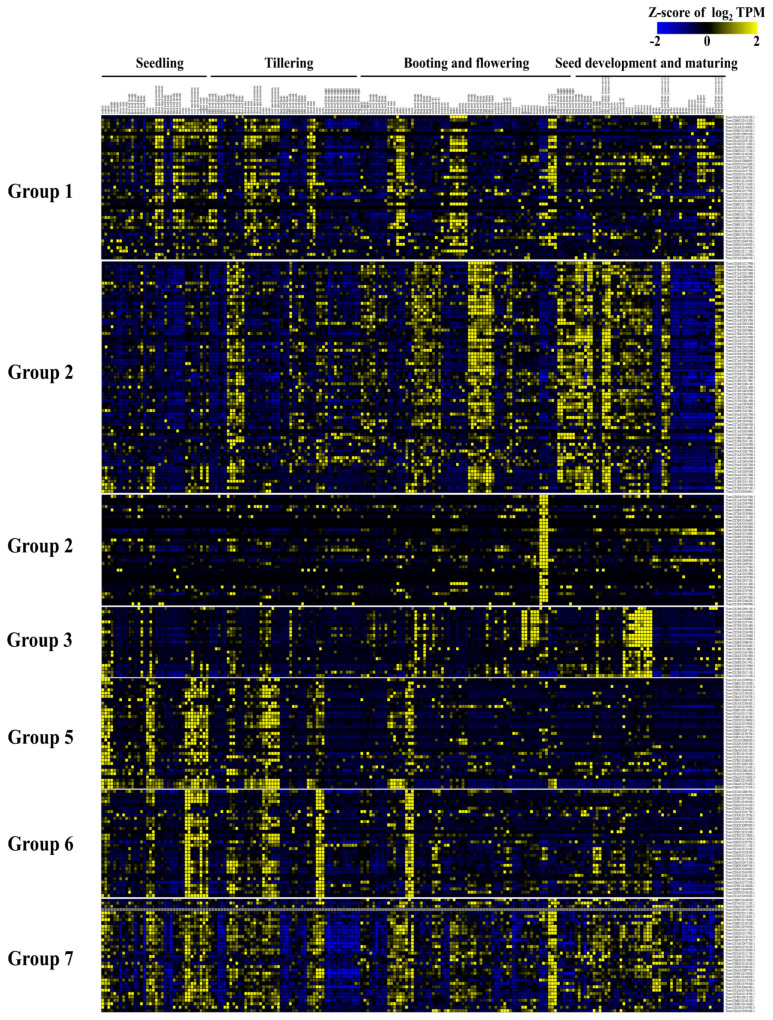
Expression profiling and classification of wheat U-box E3 genes in developmental stages. The average log_2_ transformed transcripts per million (TPM) values of U-box E3 genes at different tissue and developmental stages of RNA−sequencing samples representing 22 tissue types, from grain, root, leaf, and spike samples across multiple time points, were classified into seven sub−groups by K−means clustering method.

**Figure 8 ijms-22-02699-f008:**
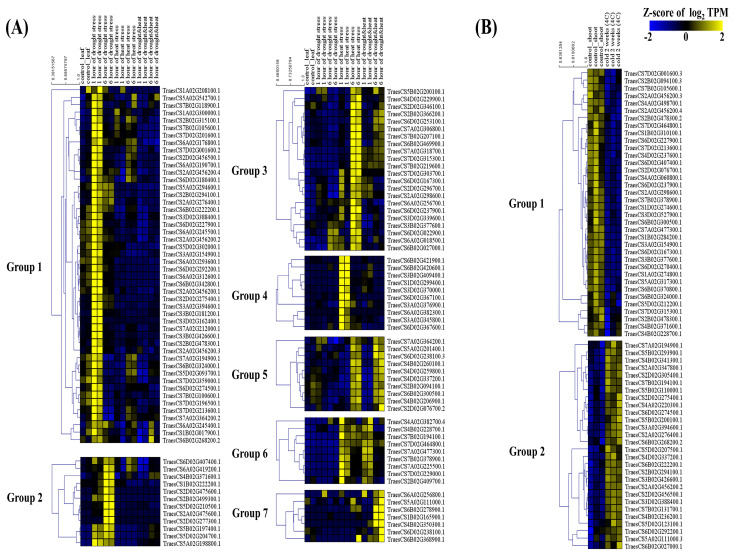
Expression profiling and classification of wheat U-box E3 genes under abiotic stress condition. The average log_2_ transformed transcripts per million (TPM) values of U-box E3 genes classified into 7 and 2 sub−groups by K−means clustering methods under (**A**) drought, heat, and combined heat and drought stress conditions, and (**B**) cold stress conditions, respectively.

**Figure 9 ijms-22-02699-f009:**
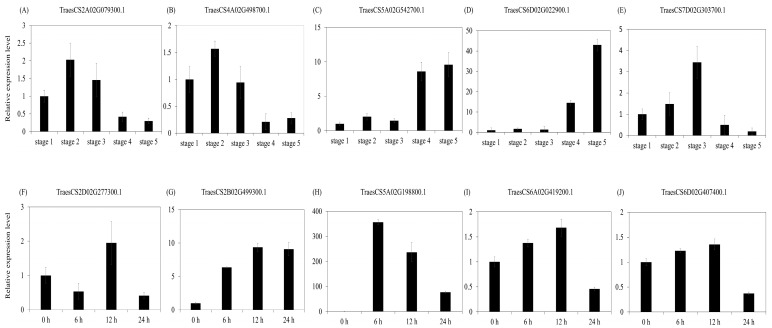
Validation of RNA-seq results by qRT-PCR in wheat at different developmental stages and drought stress conditions. (**A**–**E**) Stage 1 (leaf: three leaves emerged, Z13), stage 2 (leaf: main stem and four tillers present, Z24), stage 3 (leaf: leaf at the tip of ear just visible, Z51), stage 4 (spikelet: beginning of anthesis, Z61), and stage 5 (spikelet: early milk development, Z73) are shown. Z, Zadoks growth scale. (**F**–**J**) Seedlings in the three-leaf stage on Hoagland solution were treated with 20% PEG 6000 for simulation of drought stress; then, samples were collected 6, 12, and 24 h after treatment. All samples were normalized to the actin gene. The data are presented as the average ± standard error (*n* = 3).

## Data Availability

This research did not report any data.

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
