# Peer review of "Genome Wide Analysis of U-Box E3 Ubiquitin Ligases in Wheat (Triticum aestivum L.)"

_ijms, 2021, doi:10.3390/ijms22052699_

Round 1
Reviewer 1 Report
The manuscript entitled “Genome Wide Analysis of U-Box E3 Ubiquitin Ligases in Wheat” presents 213 U-box E3 genes in wheat and some of them showed tissue, developmental stage, and abiotic stress-specific expression patterns. I find the manuscript well written and the methods used adequately described. I have no comments to add.
Author Response
Reviewer comment:
- The manuscript entitled “Genome Wide Analysis of U-Box E3 Ubiquitin Ligases in Wheat” presents 213 U-box E3 genes in wheat and some of them showed tissue, developmental stage, and abiotic stress-specific expression patterns. I find the manuscript well written and the methods used adequately described. I have no comments to add.
Answer:
- We appreciate your valuable comments.
Reviewer 2 Report
The manuscript has a quality scientific level, it brings interesting and current knowledge and results, but its level is to some extent influenced by inconsistencies between the individual parts of the manuscript.
A significant benefit of the study is in expression profiling of U-box E3 genes under different abiotic stresses. However, it would be necessary to harmonize which types of abiotic stresses were tested. In the abstract (row 23) the authors state stresses heat, drought and their combination, in the introduction (row 76) heat, drought and salinity and in the methodological part the stress of cold is also mentioned (row 446), the results of which are declared.
Descriptions of several figures are not complete and correct. Figure 7 lacks the meaning of "Group 1" up to "Group 7". Figure 8 shows the expression profiling and classification of wheat U-box E3 genes under heat and drought abiotic stress condition and their combination after 1 and 6 hours of treatment. However, in the methodology are mentioned the time points for 6, 12 and 24 hours only for drought stress condition. In the Figure 9 it is not clear what the letters A to I mean. The information is provided only for A and B.
It would be appropriate to add the Latin name of the species to the title of the manuscript and complete the information in the sentence "Common wheat (Triticum aestivum L.) cv. Keumgang (IT 213100) developed by the National Institute of Crop Science in ...... (?).
Author Response
Reviewer comment:
- The manuscript has a quality scientific level, it brings interesting and current knowledge and results, but its level is to some extent influenced by inconsistencies between the individual parts of the manuscript. A significant benefit of the study is in expression profiling of U-box E3 genes under different abiotic stresses. However, it would be necessary to harmonize which types of abiotic stresses were tested. In the abstract (row 23) the authors state stresses heat, drought and their combination, in the introduction (row 76) heat, drought and salinity and in the methodological part the stress of cold is also mentioned (row 446), the results of which are declared.
Answer:
- As reviewer’s pointed “salinity” is not included in the types of abiotic stresses that were tested. Therefore, “salinity” is removed from the revised manuscript.
- The revised part is as follows:
In addition, expression analysis of U-box E3 genes under abiotic stress, including drought, heat, and both heat and drought, and cold condition, was conducted to provide information on U-box E3 gene expression under specific stress conditions. (Lines 25-27)
- Additionally, expression profiling of U-box E3 genes was conducted with RNA sequencing data for wheat at various developmental stages and under abiotic stresses, such as heat, drought, and cold stress conditions. (Lines 80-82)
Reviewer comment:
- Descriptions of several figures are not complete and correct. Figure 7 lacks the meaning of "Group 1" up to "Group 7". The information is provided only for A and B.
Answer:
- The figure description of Figure 7 has been improved in the revised manuscript as follows:
Figure 7. Expression profiling and classification of wheat U-box E3 genes in developmental stages. The average log2 transformed transcripts per million (TPM) values of U-box E3 genes at different tissue and developmental stages of RNA-sequencing samples representing 22 tissue types from grain, root, leaf, and spike samples across multiple time points were classified into 7 sub-groups by K-means clustering method. (Lines 255- 259)
- The figure description of Figure 8 also has been corrected in the revised manuscript as follows:
Figure 8. Expression profiling and classification of wheat U-box E3 genes under abiotic stress condition. The average log2 transformed transcripts per million (TPM) values of U-box E3 genes classified into 7 and 2 sub-groups by K-means clustering methods under (A) drought, heat, and combined heat and drought stress conditions, and (B) cold stress conditions, respectively. (Lines 294- 297)
- Additionally, the figure descriptions of Figure 5 and 6 have been improved in the revised manuscript as follows:
Figure 5. Synteny analysis and gene duplication of wheat U-box E3 genes using the BLASTP algorithm. (A) An E-value <1e-10, minimum sequence identity >90%, and bitscore >500 were used to create a synteny map among wheat, T. urartu, and A. tauschii. The green annulus on the top left represents chromosomes of A. tauschii, and the blue annulus on the top right represents chromosomes of T. urartu. Homoeologous genes in each chromosome of wheat are linked by lines with identical color. (B) Duplicated U-box E3 gene pairs identified in wheat with the alignment covered >80% of the longer gene and the identity the aligned region >90%. Duplicated gene pairs are depicted in corresponding colors and linked using lines with the corresponding color. (Lines 195-202)
Figure 6. Synteny analysis of wheat U-box E3 genes with (A) Brachypodium, (B) barley, and (C) rice. For synteny analysis of wheat against Brachypodium and rice, BLASTP was conducted with an E-value threshold of 1e-10 and sequence identity >80%. Colored lines denote syntenic regions between wheat chromosomes and other plants. The genome size of Brachypodium and rice was enlarged by a factor of 10. (Lines 224- 228)
Reviewer comment:
- Figure 8 shows the expression profiling and classification of wheat U-box E3 genes under heat and drought abiotic stress condition and their combination after 1 and 6 hours of treatment. However, in the methodology are mentioned the time points for 6, 12 and 24 hours only for drought stress condition.
Answer:
- “Heat and drought abiotic stress condition and their combination after 1 and 6 hours of treatment after 1 and 6 hours of treatment” is the methodology of RNA sequencing analysis in Figure 8A, on the other hands, “the time point for 6, 12 and 24 hours under drought stress condition” is the information of seedling samples used in qRT-PCR for the validation of RNA sequencing analysis in Figure 9F-9J. The methodology of RNA sequencing for heat and drought abiotic stress condition and their combination had been described in the lines 472- 475 of “Materials and Methods” section in the submitted manuscript.
Reviewer comment:
- In the Figure 9 it is not clear what the letters A to I mean
Answer:
- The legend of Figure 9 has been re-written for the clarity. The revised legend is as follows:
Figure 9. Validation of RNA-seq results by qRT-PCR in wheat at different developmental stages and drought stress conditions. (A- E) Stage 1 (leaf: three leaves emerged, Z13), stage 2 (leaf: main stem and four tillers present, Z24), stage 3 (leaf: leaf at the tip of ear just visible, Z51), stage 4 (spikelet: beginning of anthesis, Z61), and stage 5 (spikelet: early milk development, Z73) are shown. Z, Zadoks growth scale. (F- J) Seedlings in the three-leaf stage on Hoagland solution were treated with 20% PEG 6000 for simulation of drought stress; then samples were collected 6, 12, and 24 h after treatment. All samples were normalized to the actin gene. The data are presented as the average ± standard error (n = 3).
Reviewer comment:
- It would be appropriate to add the Latin name of the species to the title of the manuscript and complete the information in the sentence "Common wheat (Triticum aestivum L.) cv. Keumgang (IT 213100) developed by the National Institute of Crop Science in ...... (?).
Answer:
- The title of the manuscript was changed as reviewer suggested. The revised title is:
Genome Wide Analysis of U-Box E3 Ubiquitin Ligase Enzymes in Wheat (Triticum aestivum L.)
- The sentence in the lines 483- 485 also has been revised. The new sentence is:
The seeds of common wheat (Triticum aestivum L.) cv. Keumgang (IT 213100) developed by the National Institute of Crop Science was vernalized at 4 °C for 4 weeks to synchronize growth and then the wheat seedlings were transferred to soil (Sunshine Mix 1, Sun Gro Horticulture, MA, USA) pots. (Lines 483- 485)